# Exploring the Role of Bempedoic Acid in Metabolic Dysfunction Associated Steatotic Liver Disease: Actual Evidence and Future Perspectives

**DOI:** 10.3390/ijms25136938

**Published:** 2024-06-25

**Authors:** Elena Butera, Fabrizio Termite, Giorgio Esposto, Linda Galasso, Irene Mignini, Raffaele Borriello, Maria Elena Ainora, Luca Miele, Antonio Gasbarrini, Maria Assunta Zocco

**Affiliations:** 1Internal Medicine, Fondazione Policlinico Universitario “A.Gemelli” IRCCS, Università Cattolica del Sacro Cuore, Rome 00168, Italy; elena.butera23@gmail.com; 2CEMAD Digestive Disease Center, Fondazione Policlinico Universitario “A.Gemelli” IRCCS, Università Cattolica del Sacro Cuore, Rome 00168, Italy; fabrizio.termite@libero.it (F.T.); giorgio.esposto@guest.policlinicogemelli.it (G.E.); linda.galasso0817@gmail.com (L.G.); irene.mignini@guest.policlinicogemelli.it (I.M.); raffaeleborr@gmail.com (R.B.); mariaelena.ainora@policlinicogemelli.it (M.E.A.); luca.miele@unicatt.it (L.M.); antonio.gasbarrini@unicatt.it (A.G.)

**Keywords:** MASLD, bempedoic acid, dyslipidemia, ATP-citrate lyase inhibitor, ACLY, lipid-lowering therapies

## Abstract

Metabolic dysfunction-associated steatotic liver disease (MASLD) involves excessive lipid accumulation in hepatocytes, impacting global healthcare due to its high prevalence and risk of progression to severe liver conditions. Its pathogenesis involves genetic, metabolic, and inflammatory factors, with cardiovascular events as the leading cause of mortality. This review examines the role of lipid-lowering therapies in MASLD, with a particular focus on bempedoic acid, a recently approved cholesterol-lowering agent for hypercholesterolemia and high cardiovascular-risk patients. It explores its potential in liver disease by modulating lipid metabolism and inflammatory pathways based on the most recent studies available. Bempedoic acid inhibits ATP-citrate lyase, reducing cholesterol and fatty acid synthesis while activating AMP-activated protein kinase to suppress gluconeogenesis and lipogenesis. Animal studies indicate its efficacy in reducing hepatic steatosis, inflammation, and fibrosis. Bempedoic acid holds promise as a therapeutic for MASLD, offering dual benefits in lipid metabolism and inflammation. Further clinical trials are required to confirm its efficacy and safety in MASLD patients, potentially addressing the multifaceted nature of this disease.

## 1. Introduction

Metabolic dysfunction-associated steatotic liver disease (MASLD), formerly known as non-alcoholic fatty liver disease (NAFLD) and metabolic dysfunction-associated fatty liver disease (MASLD), is defined by the excessive accumulation of lipids in hepatocytes (identified by imaging or biopsy) in patients who encounter at least one cardiometabolic risk factor, in the absence of excessive alcohol consumption [1]. MASLD is a globally significant burden in both healthcare and economic settings. It is currently the most prevalent chronic liver disease worldwide, affecting over 30% of the global adult population, with prevalence peaking at 40–45% in regions such as the Middle East and South America [2]. Cardiovascular events stand as the primary cause of mortality in MASLD patients, followed by hepatic events. Currently, hepatic complications related to MASLD (liver failure and hepatocellular carcinoma—HCC) are the second most prevalent cause of liver transplantation, with projections placing it at the forefront in the coming decades [3,4]. MASLD encompasses a spectrum of histopathological conditions, spanning from simple steatosis to metabolic dysfunction-associated steatohepatitis (MASH), progressing to fibrosis and cirrhosis. Simple steatosis is a “benign” condition without signs of inflammation or fibrosis, which can evolve in approximately 20–25% of individuals into MASH [5,6], which is characterized by hepatocyte ballooning and inflammation. Around one-third of individuals with MASH experience fibrosis progression, with 20% advancing to cirrhosis [7,8]. Among patients with MASH-related cirrhosis, HCC develops with an annual incidence ranging from 0.5% to 2.6% [9,10]. HCC can also arise in non-cirrhotic MASH scenarios [11,12,13,14,15].

Despite the identification of various risk factors associated with the development of MASLD, such as metabolic syndrome, obesity, insulin resistance, dyslipidemia, and genetic variants, the reason why most patients experience benign simple steatosis while others progress to MASH remains partially elusive [16]. Recent data from scientific literature support the hypothesis that the buildup of toxic lipids within the liver, which interact with pro-inflammatory signals, induces cellular abnormalities, subsequently triggering inflammation and fibrosis [17,18,19]. Triglycerides are the main component of lipid droplets (LD) stored in hepatocytes and represent “safe” and “inert” storage molecules together with cholesterol esters [20,21]. Conversely, toxic lipids, such as free unesterified cholesterol (FC) along with free fatty acids [22], diacylglycerols [23], and ceramides [24], are toxic to various cellular processes and organelles, even with modest quantitative level increases. Some authors propose hepatic FC as the primary trigger of inflammatory response in a subgroup of MASLD patients, introducing the term “cholesterol-associated steatohepatitis” (CASH) [25]. According to this hypothesis, genetic, epigenetic, dietary, metabolic, and lifestyle factors act as co-factors leading to FC accumulation or interacting with it to advance MASH. This hypothesis finds support in the favorable outcomes achieved in MASLD/MASH with the use of lipid-lowering drugs, such as statins, ezetimibe, PCSK9 (Proprotein Convertase Subtilisin/Kexin type 9) inhibitors and peroxisome proliferator-activated receptor (PPAR) agonists. New evidence, although still in its early stages and mainly from murine models, is emerging regarding bempedoic acid (BemA), which has recently been approved as a cholesterol-lowering agent in patients with hypercholesterolemia and high cardiovascular risk who have an inadequate response to or are intolerant of conventional statin therapy. This review aims to delineate bempedoic acid’s role in MASLD/MASH, delving into the molecular pathogenetic mechanisms involved.

## 2. Role of Cholesterol in the Pathogenesis of MASLD and MASH

Human and experimental studies on various animal species highlighted that the addition of dietary cholesterol accelerates the progression of MASLD to MASH-cirrhosis.

For instance, an analysis of a multiethnic cohort study involving over 215,000 older adults from California and Hawaii found a positive connection between dietary cholesterol intake and the development of MASLD-cirrhosis (OR = 1.52; 95%, CI = 1.15–2.01; *p* value = 0.002) [26]. Another study conducted on the U.S. population demonstrated a significant correlation between dietary cholesterol intake (excluding overall fat consumption) and the incidence of cirrhosis across all etiologies of liver disease [27]. This pathogenetic correlation has been further confirmed by the identification of genetic variants predisposing to the development of MASLD (e.g., patatin-like phospholipase domain-containing protein 3 PNPLA3, transmembrane 6 superfamily member 2 TM6SF2, Apoliprotein B) which are also involved in hepatic cholesterol metabolism processes [28,29,30].

Cholesterol, a vital lipid molecule in animal cells, plays multiple biological roles. It guarantees the barrier function between the cell and its environment, regulates membrane fluidity and forms lipid rafts that concentrate signaling molecules [31]. Additionally, cholesterol serves as a crucial metabolic precursor for vitamins, bile acids, and steroid hormones [32]. However, excessive accumulation of cholesterol in the bloodstream, a condition known as hypercholesterolemia, can lead to harmful complications like MASLD/MASH and atherosclerosis [25,33]. Impaired cholesterol homeostasis affects several constituent cell types of the liver parenchyma, including hepatocytes (cells deputed to catabolic and anabolic functions, as well as the cells that accumulate the greatest amount of cholesterol in the form of LD), hepatic stellate cells—HSC—(primarily responsible for the deposition of fibrous material), Kuppfer cells—KC—(resident macrophages of the liver that, once activated, produce pro-inflammatory cytokines capable of activating HSC) and sinusoidal endothelial cells [34]. Cholesterol also alters the function of organelles such as mitochondria and endoplasmic reticulum (ER) [34,35,36]. Hence, the liver’s excessive buildup of FC initiates various pathogenetic mechanisms, all of which share a common feature: the activation of inflammatory pathways that potentially promote steatohepatitis [25].

KC uptake cholesterol through receptor-mediated endocytosis of circulating lipoproteins, including cholesterol-rich LDL [37]. While non-modified LDL uptake is subject to feedback inhibition (e.g., suppression of the transcription of LDL receptor gene) to prevent cellular cholesterol overload, modified LDL (e.g., oxidized LDL) are taken up by scavenger receptors, devoid of any feedback mechanism, leading to unregulated uptake and foam cell formation [37,38]. Foam cells produce cytokines with fibrogenic properties, including tumor necrosis factor-alpha (TNF-α) and transforming growth factor beta (TGF-β), which can stimulate HSC in the deposition of fibrotic matrix [38]. HSC can also be directly activated by a signal molecule known as Indian Hedgehog (Ihh), released into the extracellular environment by FC-overloaded hepatocytes [39]. Accumulated FC in hepatocytes blocks proteasomal degradation of TAZ (a transcriptional regulatory factor), which translocate into the nucleus and induces Ihh transcription [39]. However, recent evidence suggests that HSC may also be directly activated by cholesterol accumulation through an independent pathway represented by toll-like receptor-4 (TLR-4) [40].

Moreover, recent findings reveal the presence of cholesterol crystals within the LD of fatty hepatocytes in individuals diagnosed with MASH induced by a high cholesterol diet [24]. Interestingly, these crystals were absent in patients with simple steatosis despite exhibiting a comparable level of fat accumulation, characterized by similar LD numbers and sizes. Some studies speculate that the accumulation of cholesterol crystals in hepatocytes prompts NLPR3 inflammasome activation and triggers pyroptosis (a mechanism of programmed cell death) [41]. Remnant cholesterol crystals released from dead hepatocytes are then phagocytosed by KC. Similarly, KC activate the NLPR3 inflammasome pathway, but, in this specific scenario, with a different outcome: caspase 1 activation occurs, leading to the cleavage of pro-interleukin-1β (IL-1β) into its biologically active form IL-1β, triggering inflammatory processes typical of innate immunity [42].

Excess hepatocyte FC also results in altered composition of organelle membranes. Indeed, an abnormal FC/phospholipid ratio results in important dysregulation of metabolic activities. FC accumulation within mitochondria disrupts the function of the 2-Oxoglutarate dehydrogenase complex, leading to depletion of the mitochondrial glutathione pool [34]. This depletion results in the generation of reactive oxygen species (ROS), lipid peroxidation, release of cytochrome C, and initiation of cell apoptosis. Elevated FC in the ER membrane alters sarco-/endoplasmic reticulum Ca^2+^-ATPase (SERCA) function, leading to reduced luminal calcium levels and increased unfolded proteins, exacerbating ER stress. The unfolded protein response (UPR) upregulates key enzymes to alleviate ER stress, but chronic ER stress can activate the NLPR3 inflammasome, with all that comes with it (IL-1β production in KC, pyroptosis in hepatocytes) [35,36].

The above-mentioned pathways are summarized in Figure 1. 

## 3. Lipid-Lowering Drugs and MASLD

In MASLD patients, clinicians often address comorbidities such as type 2 diabetes mellitus (T2DM), hypertension, obesity, and atherogenic dyslipidemia, which are commonly associated with metabolic syndrome. Statins, renowned for their lipid-lowering effects, also possess additional properties such as antioxidative and anti-inflammatory effects, promotion of new blood vessel formation, and enhancement of endothelial functions [43]. Statins work by reducing cellular cholesterol content through selective inhibition of the enzyme HMG-CoA reductase, which limits cholesterol biosynthesis and lowers hepatic cholesterol levels. This results in increased expression of LDL receptors (LDL-R) on liver cell membranes, enhancing the clearance of circulating LDL-cholesterol particles from the blood. In certain patients, such as those with combined hyperlipidemia, statin therapy decreases the hepatic production rate of apo B100-containing lipoproteins, ultimately leading to a reduction in both cholesterol and triglyceride levels [44].

Along with their lipid-lowering effect, statins also exhibit pleiotropic properties, including antioxidative and anti-inflammatory effects, neoangiogenesis, and improvement of endothelial functions. Emerging evidence suggests that statin therapy in MASLD patients is linked to notable improvements in liver conditions such as steatosis, inflammation, and even fibrosis [45,46,47]. For instance, a recent observational study of 11,593,409 individuals from the National Health Information Database of the Republic of Korea revealed that statin use was associated with reduced MASLD risk (adjusted odds ratio 0.66, 95% confidence interval 0.65–0.67) and diminished risk of significant liver fibrosis, as assessed indirectly by the BARD score, even after adjusting for various metabolic risk factors (adjusted odds ratio 0.43, 95% confidence interval 0.42–0.44) [46].

Moreover, MASLD patients face elevated risks of cardiovascular morbidity and mortality. In a retrospective analysis of the Greek Atorvastatin and Coronary Heart Disease Evaluation (GREACE) study, which included 437 patients with moderately abnormal liver tests attributed to MASLD (227 of whom received statin treatment and 210 did not), Athyros et al. [48] found that statin-treated MASLD patients experienced significantly reduced cardiovascular morbidity without notable liver-related adverse events. 

Animal studies have also elucidated potential mechanisms underlying statins’ beneficial effects on liver histology in MASLD. In experimental non-alcoholic steatohepatitis, statins influence paracrine signaling between hepatocytes and HSC, inhibiting HSC activation and subsequent fibrosis processes, particularly through pathways like RhoA/Rho-kinase. Additionally, in other models like angiotensin-II-induced liver fibrosis, statins have demonstrated the ability to reduce fibrosis by suppressing inflammatory activity [49].

Ezetimibe functions as a lipid-lowering agent by inhibiting cholesterol absorption in the intestines. Its mechanism involves blocking the Niemann–Pick C1-like 1 (NPC1L1) protein, a crucial mediator of cholesterol absorption, present in both gastrointestinal tract epithelial cells and hepatocytes. A meta-analysis of six studies [49], comprising two randomized controlled trials (RCTs) and four single-arm trials involving 273 MASLD patients with and without T2DM, revealed that ezetimibe significantly lowered serum liver enzyme levels (AST, ALT, γ-GTP) and improved hepatic steatosis and hepatocyte ballooning. However, this study did not find evidence of ezetimibe improving liver fibrosis.

Proprotein convertase subtilisin kexin type-9 (PCSK-9) plays a pivotal role in cholesterol homeostasis by inhibiting the LDL receptor (LDL-R) pathway. When secreted by hepatocytes, PCSK-9 binds to the LDL-R’s extracellular EGF(A) domain, leading to its lysosomal degradation. Increasing data indicate that elevated intrahepatic or circulating PCSK-9 levels contribute to increased lipid storage in muscles and liver, adipose energy storage, hepatic fatty acids, and triglycerides storage, thereby promoting MASLD development [50]. Early evidence suggests that antisense particles targeting PCSK-9 mRNA or anti-PCSK-9 antibodies, capable of reducing circulating PCSK-9 levels, may improve MASLD independent of LDL cholesterol reduction [51].

Another class of hypolipidemic drugs with a wide entry margin among future treatment options for MASLD are PPAR agonists. Fenofibrate, the most widely used PPARα agonist, enhances fatty acid β-oxidation and boosts lipoprotein lipase activity, thereby reducing plasma triglycerides and VLDL while increasing HDL cholesterol levels [52,53]. Although six clinical trials investigating fibrates for MASLD/MASH did not demonstrate significant therapeutic benefits, newer drugs in this class show potential in ongoing clinical trials, such as pemafibrate (PPARα-selective agonist) [54]. Experimental models indicate that pemafibrate improves liver function tests and histological features of fatty liver, including ballooning, inflammation, and fibrosis, by upregulating genes involved in β-oxidation and lipid transport and regulating energy metabolism through UCP3 gene induction [55,56,57,58]. Pemafibrate shows potential due to its liver-based metabolism, which minimizes renal dysfunction risks and consistently reduces serum liver enzymes like ALT, ALP, γ-GT, and bilirubin, especially in patients with elevated liver function values [59]. A phase 2 study with 118 MASH patients demonstrated that while pemafibrate did not significantly alter liver fat content, it significantly reduced liver stiffness, ALT, and LDL-C levels over 72 weeks, indicating its role in reducing liver fibrosis and improving liver function [60]. Additionally, other PPAR agonists are being investigated for the treatment of MASLD in phase 2 (saroglitazar—PPARα/γ dual agonist and chiglitazar—PPARα/δ/γ pan agonist) and phase 3 studies (lanifibranor—PPARα/δ/γ pan agonist) [54].

## 4. Pharmacokinetics of Bempedoic Acid

Cicero et al. examined the pharmacokinetics of bempedoic acid (BemA) in the treatment of hypercholesterolemia [61]. The mean half-life of BemA was reported to range from 16 to 33 h which supports once daily dosing [62]. The advised daily dosage is a 180 mg tablet, taken orally with or without food. After oral administration, BemA is well absorbed by the enteric mucosa; it moderately distributes into body fluids (volume of distribution approximately 18 L) and does not penetrate erythrocytes. Plasma levels (AUC and Cmax) vary linearly with the dose over a wide range (60–220 mg). Biotransformation primarily occurs in the liver without the involvement of cytochrome P450 (CYP).

A significant portion of the dose is converted by aldoketoreductase into the derivative ESP15228, which has pharmacological activity comparable to the parent compound but achieves plasma levels (AUC) over five times lower (around 18% at steady state), thus contributing marginally to the overall pharmacological effect. Both compounds are eventually converted by UGT2B7 into inactive glucuronides, which are excreted predominantly in the urine and, to a lesser extent, in the feces, along with small amounts of unchanged bempedoic acid [63].

BemA and its active form, bempedoyl-CoA, are predominantly eliminated (70%) through conjugation with glucuronic acid and renal excretion, while the liver accounts for about 30% of the clearance [64]. In patients with mild or moderate hepatic cirrhosis (Child–Pugh classes A and B), the AUC decreases by 22% and 16%, respectively, with similar reductions (−23% and −36%) observed for the active metabolite ESP15228. Patients with mild to moderate hepatic impairment (Child–Pugh class A and B) do not require dosage adjustments. However, bempedoic acid should be avoided in patients with severe liver disease (Child–Pugh class C) [62]. Renal insufficiency leads to a proportional increase in plasma bempedoic acid levels; for CrCl values of 60–89, 30–59, and <30 mL/min, the drug’s AUC increases by 40–50%, 90–130%, and 140%, respectively. Given that generally, patients with chronic kidney disease well tolerate BemA, no dose adjustments are currently needed for mild or moderate renal impairment. Bempedoic acid has not been adequately studied in patients with severe renal impairment (eGFR < 30 mL/min/1.73 m^2^) or in those with end-stage renal disease on dialysis [65].

The safety of bempedoic acid in long-term use appears satisfactory from phase 3 clinical trials: CLEAR (Cholesterol Lowering via BEmpedoic Acid, an ACL-inhibiting Regimen) Tranquility, CLEAR Serenity, CLEAR Wisdom, and CLEAR Harmony [66,67,68,69]. Bempedoic acid was approved by the FDA for reducing LDL-C in February 2020 [69], and the EMA recommended approval in January 2020 [70].

## 5. Bempedoic Acid and MASLD

Bempedoic acid is a pro-drug that is converted in the liver into the active form, bempedoyl-CoA, by the very long-chain acyl-CoA synthetase-1 (ACSVL1), an enzyme present in the hepatocyte but completely absent in skeletal muscle [71]. The effect of BemA on cholesterol synthesis is similar to that of statins but acts at a higher level on the metabolic synthesis pathway. ACSVL1 is not expressed in skeletal muscle, impeding the conversion of BemA into its active form. Therefore, in skeletal muscle, BemA should not promote toxicity associated with cholesterol synthesis inhibition. 

The beneficial effects of BemA on metabolism and inflammation are closely linked to the modulation of two key enzymes that regulate lipid, carbohydrate, and energy metabolism [71,72,73]. BemA exerts its pharmacological effects through multiple mechanisms. First, it inhibits hepatic ATP-citrate lyase (ACLY), a key enzyme involved in regulating lipid synthesis and gluconeogenesis. By inhibiting ACLY, BemA interferes with the conversion of citric acid from the mitochondrial Krebs cycle into oxaloacetate and acetyl-CoA in the cytoplasm. This leads to reduced gluconeogenesis and hepatic glucose production due to decreased availability of oxaloacetate. Additionally, acetyl-CoA serves as a precursor for both lipogenesis and sterol synthesis. Lipogenesis involves the formation of malonyl-CoA by acetyl-CoA carboxylase (ACC), followed by the synthesis of fatty acids and triglycerides. Similarly, cholesterol synthesis occurs through the formation of 3-hydroxy 3-methylglutaryl-CoA (HMG-CoA) and mevalonic acid catalyzed by HMG-CoA reductase (HMGR), which is targeted by statins. Inhibition of hepatic ACLY by BemA results in the suppression of cholesterol synthesis. This triggers a compensatory increase in membrane receptors for LDL, thereby enhancing LDL clearance from the bloodstream [74]. Furthermore, bempedoic-free acid has been shown to upregulate the AMP-activated protein kinase (AMPK), a crucial kinase that plays a central role in regulating gluconeogenesis. AMPK upregulation leads to the downregulation of key rate-limiting enzymes involved in gluconeogenesis, such as glucose-6-phosphatase (G6Pase) and phosphoenolpyruvate carboxykinase (PEPCK). G6Pase hydrolyzes glucose 6-phosphate to generate free glucose for export from the cell, while PEPCK catalyzes the conversion of oxaloacetate into phosphoenolpyruvate, initiating gluconeogenesis. Additionally, ACLY inhibition by BemA acid further reduces glucose production by limiting oxaloacetate availability for gluconeogenesis [74]. Moreover, AMPK upregulation inhibits the rate-limiting enzymes of fatty acid and cholesterol synthesis pathways, namely acetyl-CoA carboxylase (ACC) and HMG-CoA reductase (HMGR), respectively. Thus, BemA targets both glucose and lipid metabolism pathways, resulting in decreased liver fatty acid and cholesterol synthesis. The combined effects of BemA on these molecular pathways lead to significant reductions in LDL-C synthesis and systemic inflammation, with additional benefits observed in metabolic syndrome and diabetes management [75]. The described pathways are shown in Figure 2.

Furthermore, the beneficial effects of BemA on cardiovascular outcomes are being assessed in the CLEAR Outcomes study (ClinicalTrials.gov Identifier: NCT02993406). Among patients for whom primary or secondary prevention of cardiovascular disease is clinically indicated but who were unable or unwilling to take guideline-recommended doses of statins, the risk of a primary end-point event (death from cardiovascular causes, nonfatal myocardial infarction, nonfatal stroke, or coronary revascularization) was significantly lower by 13% with BemA 180 mg once daily than with placebo after a median of 40.6 months of follow-up, with an absolute between-group difference in incidence of 1.6 percentage points [76].

In addition to its metabolic actions, BemA exhibits potent systemic anti-inflammatory effects, as evidenced by its ability to significantly reduce circulating high-sensitivity C-reactive protein (hsCRP) levels in vivo. In LDLR−/− mice fed with a high-fat diet, supplementation with BemA resulted in reductions in plasma and tissue lipid levels, attenuation of pro-inflammatory gene expression, and suppression of cholesteryl ester accumulation in the aortic wall, ultimately preventing the development of atherosclerotic plaques [77]. Mendelian randomization studies have shown that genetic variants mimicking the effects of ACLY inhibition are linked to decreased levels of serum biomarkers associated with cardiovascular risk, including triglycerides and LDL cholesterol, as well as reductions in MASLD, body fat, and T2DM [78]. These findings align with the observed inter-individual variability in ACLY expression, which could influence gluconeogenesis efficiency and the effectiveness of BemA. Consequently, the pharmacogenetics of ACLY inhibitors presents a promising avenue for personalized medicine.

Animal studies [75,77,79,80] have revealed that elevated hepatic de novo lipogenesis contributes significantly to MASLD, and inhibition of ATP-citrate lyase (ACLY), a key enzyme in lipogenesis, has been shown to effectively control hepatic steatosis. In animal models, BemA, by inhibiting ACLY, has demonstrated efficacy in reducing hepatic steatosis and preventing the progression of MASLD to MASH by impairing the activation and proliferation of HSC, thereby reducing fibrosis. While animal models have provided valuable insights, clinical studies are essential to confirm the efficacy of BemA in humans. 

MASH induced by a long-term high-fat diet (HFD) in animal models closely mimics the characteristics of human MASH and, hence, is used by investigators as a model system for studying the mechanism of action of new drugs [80]. Overall, BemA has shown promise in reducing HFD-induced MASH by inhibiting body weight gain, improving glycemic control, reducing hepatic triglycerides and total cholesterol levels, modulating inflammatory and fibrotic genes, and enhancing the NAS (NAFLD activity score). These findings align with existing literature on the inhibition of hepatic de novo lipogenesis and the protective effects against hepatic steatosis [80].

Another study [81] used female Sprague–Dawley rats fed with a high-fat diet supplemented with fructose (HFHFr) as an animal model of liver steatosis; the authors administered BemA during the last month to a subgroup of rats and then analyzed various parameters including zoometric, plasmatic, gene and protein expression, as well as PPAR-PPRE binding activity. The results indicated a likelihood of developing hepatic steatosis and hypertriglyceridemia in the interventional model. Interestingly, despite increased caloric intake, no significant weight gain was observed, describing a new dietary model of fatty liver without the concomitant induction of liver inflammation, obesity, and clear signs of whole-body insulin resistance. Notably, treatment with BemA strongly reversed hepatic triglyceride deposition, reduced hepatic steatosis, and induced hepatocyte hypertrophy. These changes were accompanied by a significant increase in liver weight without affecting markers of inflammation, oxidative stress, or endoplasmic reticulum stress. The authors also determined plasma analytes, liver histology, adiposity, the expression of key genes controlling fatty acid metabolism and PPAR agonism [82]. They suggest three novel mechanisms that could account for the anti-steatotic effect sorted by bempedoic acid: (1) reduction of liver ketohexokinase, leading to lower fructose intake and reduced de novo lipogenesis; (2) increased expression of PNPLA3, a protein related to the export of liver triglycerides to blood; and (3) PPARα agonist activity, leading to increased hepatic fatty acid β-oxidation. 

Another mechanism that is impaired in liver disorders is the production of hydrogen sulfide (H2S), which is involved in several diverse physiological and pathophysiological processes, including obesity, hypertension, T2DM, and MASLD. 

Upon these bases, Roglans et al. [83] studied whether the HFHFr dietary intervention in female rats was able to induce a reduction in liver H2S production and, if so, to test whether BemA administration could reverse these changes. In this investigation, liver samples were used to assess both total and enzymatic H2S production. Additionally, the expression levels of three key enzymes involved in the H2S production pathway through transsulfuration, namely cystathionine β-synthase (CBS), cystathionine γ-lyase (CSE), and 3-mercaptopyruvate sulfurtransferase (3MST), were analyzed, as well as the expression and activity of farnesoid X receptor (FXR), a transcription factor implicated in regulating CSE expression. The findings indicate that the HFHFr diet led to a reduction in both total and enzymatic liver H2S production, primarily attributable to decreased expression levels of CBS and CSE. Remarkably, treatment with BemA reversed this effect, resulting in restored H2S production and increased expression of CBS and CSE. This suggests the involvement of FXR transcriptional activity and the mTORC1/S6K1/PGC1α pathway in mediating the effects of BemA on H2S production.

Another potential pathway implicated in MASLD pathogenesis is hepatic lipogenesis mediated by the SLC13A5/SLC25A1 ACLY-dependent signaling pathway, as described by Sun et al. [84]. Silencing or pharmacological inhibition of SLC25A1/ACLY in murine primary hepatocytes and HepG2 cells resulted in a notable increase in SLC13A5 transcription and activity. This upregulation of SLC13A5 activity led to heightened lipogenesis, indicating a compensatory response when the SLC25A1/ACLY pathway was inhibited. However, treatment with BemA effectively countered this upregulation, reducing lipid accumulation and improving various MASLD biomarkers. The disease-modifying effects of BemA were further validated in MASLD mice. Mechanistic investigations unveiled that BemA could reverse the heightened transcription levels of SLC13A5 and ACLY, along with the consequent lipogenesis induced by PXR activation, both in vitro and in vivo. Crucially, this effect was attenuated upon the knockdown of PXR, underscoring the involvement of the hepatic PXR-SLC13A5/ACLY signaling axis in mediating BemA’s action [85].

Finally, as half of individuals with MASLD also have a related metabolic comorbidity (including 69% with hyperlipidemia, 51% with obesity, 39% with hypertension, 22% with type 2 diabetes, and 42% with metabolic syndrome) [16], combination therapies targeting different pathways of MASH pathophysiology could lead to better efficacy and more tolerability. In their study, Desjardins et al. [82] demonstrated that the combination of BemA and the GLP-1 agonist liraglutide effectively reduces liver steatosis, hepatocellular ballooning, and hepatic fibrosis in a mouse model of MASH. Their analysis of liver RNA revealed an additive downregulation of pathways associated with MASH resolution. Furthermore, they observed reductions in the expression of clinically prognostic genes compared to clinical MASH samples, along with a gene signature profile suggestive of fibrosis resolution.

The characteristics of the described studies can be found in Table 1. 

Regarding clinical trials, we consulted Clinical-Trials.gov on 19 June 2024, and retrieved only one ongoing trial on BemA and MASLD [86]. This randomized clinical trial (NCT06035874) aims to evaluate the effects of BemA on liver fat in patients with MASLD and T2DM. Patients are randomized between BemA 180 mg tablet once a day for 24 weeks (Bemp group) and standard treatment (control group). Hepatic steatosis is measured via Magnetic Resonance Imaging Proton Density Fat Fraction (MRI-PDFF), while controlled attenuation parameter (CAP) and liver stiffness measurement (LSM) are measured by Fibroscan. The study started in October 2023, and it will reach its conclusion in December 2024 with an estimated enrollment of 100 patients. 

## 6. Discussion

The pathogenesis of MASLD involves complex interactions among various factors, including genetic predisposition, metabolic syndrome, and dietary influences. Dysregulated cholesterol metabolism, characterized by the accumulation of toxic lipids like FC, plays a pivotal role in initiating inflammation and fibrosis within the liver. Understanding these molecular mechanisms provides valuable insights into potential therapeutic targets. Currently, the only drug available to treat non-cirrhotic MASH is Resmetirom, recently approved by the FDA. It is a selective thyroid hormone receptor-β agonist designed to improve MASH by increasing hepatic fat metabolism and reducing lipotoxicity [87]. Alongside this new drug, the only available and efficacious treatment for this condition and its advanced stages remains lifestyle changes to promote weight loss. 

Moreover, a current challenge in the realm of MASLD is the absence of dependable and non-invasive end-points for MASH in clinical trials. Furthermore, while the resolution of MASH and/or improvement in fibrosis are commonly accepted end-points, histological evaluation through liver biopsy is frequently suboptimal and invariably invasive. Statins and PCSK9 inhibitors have shown promise in managing dyslipidemia associated with MASLD. Statins exhibit pleiotropic effects beyond lipid lowering, including anti-inflammatory and anti-fibrotic properties. Ezetimibe, PCSK9 inhibitors, and PPAR agonists offer additional avenues for lipid management in MASLD patients. 

BemA, a novel cholesterol-lowering agent, emerges as a promising therapeutic agent in the management of metabolic dysfunctions underlying MASLD. By inhibiting hepatic ACLY, BemA disrupts lipogenesis and gluconeogenesis pathways, leading to reduced hepatic lipid accumulation. Moreover, BemA’s modulation of AMPK activity further contributes to its metabolic benefits [74]. 

The findings illustrated above from preclinical studies [80,81,82,83,84,85] in animal models suggest that BemA holds considerable potential for halting the progression of MASLD to MASH and mitigating associated metabolic complications. Moreover, BemA could be useful in addition to statins for MASLD patients that do not reach LDL targets and in which the addition of fibrates could lead to myopathies. Indeed, the selective effect of BemA on liver enzymes could come into aid in MASLD patients with statin induced myopathies. 

Based on these premises, further investigations are warranted to fully ascertain BemA’s clinical utility, particularly through well-designed clinical trials in human subjects. The first results on a human cohort will soon be available, as the ongoing randomized trial NCT06035874 is expected to reach completion in December 2024. Future trials should not only evaluate the efficacy of BemA in improving histological features of MASH but also assess its impact on relevant clinical outcomes such as liver function, cardiovascular risk factors, and overall mortality. 

Additionally, long-term safety assessments and comparisons with existing therapeutic modalities will be essential for establishing BemA’s place in the armamentarium against MASH.

## 7. Conclusions

MASLD represents a multifaceted disorder with significant implications for global public health. Current therapeutic strategies primarily focus on lifestyle interventions and the management of metabolic comorbidities. However, emerging evidence suggests the potential of pharmacological agents like statins, ezetimibe, PCSK9 inhibitors, PPAR agonists, and bempedoic acid in addressing the underlying metabolic abnormalities associated with MASLD.

Further research, particularly clinical trials evaluating the efficacy and safety of these agents in MASLD patients, is warranted. Additionally, considering the multiple pathways implicated in MASH pathogenesis, as well as the single response from single-agent therapies, it is reasonable to assume that a combination of different therapies might be more appropriate to treat MASH. Although combination therapies may increase the difficulty of trials’ design and present more challenges in patient recruitment/retention, these efforts may improve the benefit/risk ratio compared with monotherapies.

As we strive to unravel the complexities of MASLD, an integrated approach of pharmacotherapy, lifestyle interventions, and personalized medicine strategies will be crucial in fighting this growing global health epidemic. In this setting, the promising preclinical data on BemA offer hope in addition to the recently approved Resmetiron, as it aims at different molecular pathways. Therefore, further research endeavors and studies on human cohorts are imperative to fully elucidate the therapeutic potential of BemA in the management of MASH and related metabolic disorders.

## Figures and Tables

**Figure 1 ijms-25-06938-f001:**
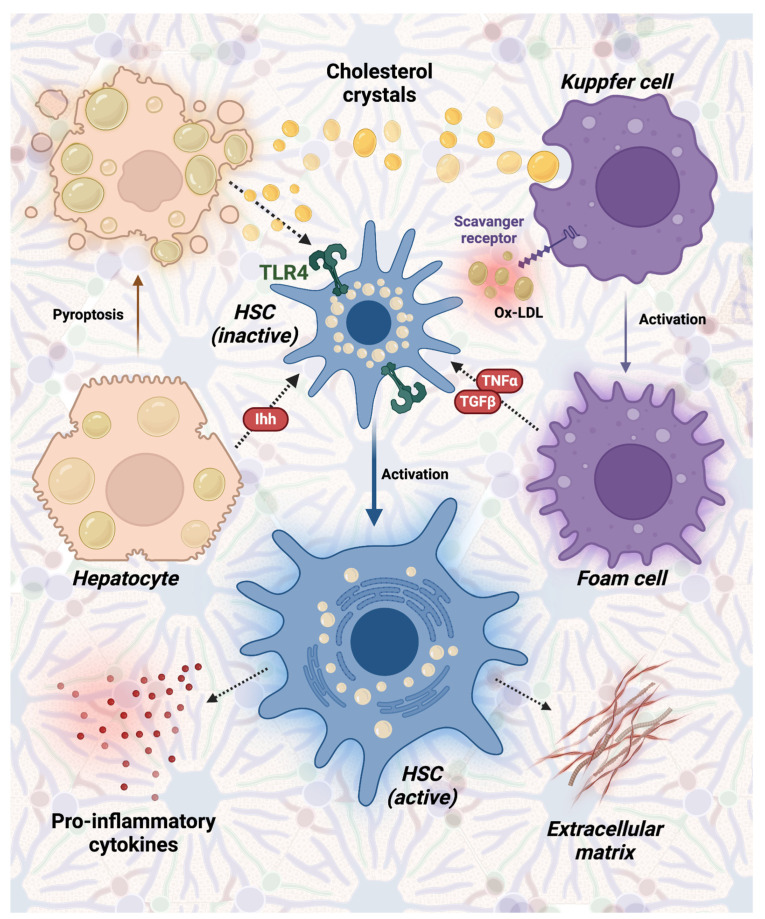
The pathogenic role of cholesterol in the development of MASH. Ox-LDL is taken up by Kuppfer cells’ scavenger receptors, leading to foam cell formation. Moreover, Kupffer cells can be activated by phagocytosing remnant cholesterol crystals released into the extracellular environment by hepatocytes undergoing pyroptosis. Foam cells secrete cytokines such as TNF-α and TGF-β, stimulating HSC to deposit extracellular matrix and pro-inflammatory cytokines. HSC can also be directly activated by the interaction between free cholesterol and TLR-4. Additionally, cholesterol-overloaded hepatocytes release Ihh, activating HSC. *Abbreviations: HSC: hepatic stellate cells, Ihh: Indian Hedgehog protein, Ox-LDL: oxidated-low-density lipoproteins, TGF-β: transforming growth factor β, TNF-α: tumor necrosis factor α, TLR4: toll-like receptor 4;* MASH, metabolic dysfunction-associated steatohepatitis.

**Figure 2 ijms-25-06938-f002:**
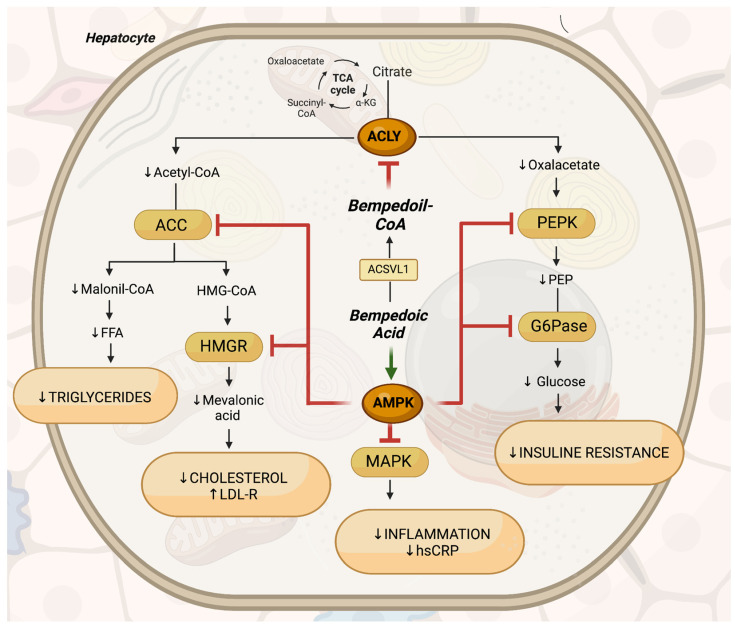
Bempedoic acid mechanism of action. BemA is a pro-drug that is metabolized into its active form, bempedoyl-CoA, within hepatocytes by ACSVL1. Bempedoyl-CoA inhibits ACLY, an enzyme critical for producing acetyl-CoA and oxaloacetate from citric acid in the mitochondrial TCA cycle. This inhibition reduces levels of acetyl-CoA and oxaloacetate, thereby affecting fatty acid and cholesterol synthesis, as well as gluconeogenesis. Additionally, BemA activates the AMPK pathway, leading to the downregulation of key enzymes involved in lipid synthesis (ACC and HMGR) and glucose production (PEPCK and G6Pase). In the liver, the combined inhibition of ACLY and upregulation of AMPK by BemA decreases cholesterol, free fatty acid (FFA), and glucose synthesis by reducing precursor availability and downregulating key enzymatic activities. This mechanism may improve conditions such as dyslipidemia and insulin resistance. Furthermore, BemA’s upregulation of AMPK downregulates the MAPK pathway, resulting in decreased inflammation and hsCRP levels, potentially aiding in the prevention and treatment of MASH. Abbreviations: α-KG: alpha-ketoglutarate, ACC: acetyl-CoA carboxylase, ACLY: adenosine-triphosphate citrate lyase, ACSVL1: very long-chain acyl-CoA synthetase-1, AMPK: adenosine monophosphate-activated protein kinase, BemA: bempedoic acid, FFA: free fatty acids, G6Pase: glucose 6-phosphatase, HMG-CoA: hydroxymethylglutaryl-CoA, HMGR: 3-hydroxy-3-methylglutaryl-CoA reductase, hsCRP: high-sensitivity C-reactive protein, LDL-R: low-density lipoprotein receptor, MAPK: mitogen-activated protein kinases, PEP: phosphoenolpyruvate, PEPK: phosphoenolpyruvate carboxykinase, TCA: tricarboxylic acid cycle.

**Table 1 ijms-25-06938-t001:** Characteristics of the included studies.

Author, Year	Type of Study	Drug	Aim	Bempedoic Acid Effects
Bentanachs R. et al., 2022 [81]	Translational study on murine models	Bempedoic Acid	Bempedoic Acid as a treatment for steatohepatitis and/or hypertriglyceridemia in the initial phases of MASLD	- Reduction in body weight - Reduction in liver triglycerides by 51%, reduction in the synthesis of cholesterol and fatty acids - Hypertrophy of hepatocytes through activation of the PPAR receptor - Reverse of steatohepatitis through a mechanism that would involve, at least partially, the direct activation of the nuclear receptor PPAR
Velázquez AM et al., 2022 [82]	Translational study on murine models	Bempedoic Acid	Bempedoic acid as a therapy for MASLD	- Reduction in body weight - Reduction of KHK expression - Reduction of liver triglycerides by 56%, reduction in the synthesis of cholesterol and fatty acids - PPAR agonist role
Roglans N. et al., 2022 [83]	Translational study on murine models	Bempedoic Acid	Role of bempedoic acid on H2S production	- Restoration of the enzymatic production of liver H2S through the expression of CBS and CSE
Sanjay K.V. et al., 2021 [80]	Translational study on murine models	Bempedoic Acid	Bempedoic acid as a therapy for MASLD	- Reduction in body weight - Improvement of glycemic control and lipid profile - Modulation of inflammation through the downregulation of the expression of Timp-1 and Col-1a1 - Reduction of fibrosis through the reduction of Mcp-1 expression - Reduction of NAS score
Sun Q. et al., 2023 [84]	Translational study on murine models	Bempedoic Acid	Hepatic lipogenesis among SLC13A5, SLC25A1, and ACLY and the therapeutic potential of bempedoic acid in MASLD	- Reverse of the elevated transcription levels of the SLC13A5 gene and ACLY and the subsequent lipogenesis induced by PXR activation - Suppression of the hepatic PXR-SLC13A5/ACLY signaling axis
Desjardins E.M. et al., 2023 [85]	Translational study on murine models	Bempedoic Acid	Effects of bempedoic acid combined with GLP1 receptor agonist in MASLD	- Reduction of body weight, adiposity, glucose intolerance, insulin resistance, and serum cholesterol - Reduction of hepatocellular ballooning scores - Reduction of lobular inflammation scores - Reduction of NAS score - Reduction of ALT, AST, and CRP - Downregulation of fibrosis-related molecular pathways

MASLD, metabolic dysfunction-associated steatotic liver disease; PPAR, peroxisome proliferator-activated receptor; KHK, ketohexokinase; H2S, hydrogen sulfide; CBS, cystathionine-β-synthetase; CSE, cystathionine-γ-lyase; NAS, NAFLD activity score; ACLY, ATP citrate lyase; PXR, pregnane X receptor; GLP1, glucagon-like peptide 1; AST, aspartate transaminase; ALT; alanine aminotransferase; Timp-1, metallopeptidase inhibitor 1; Col-1a1, alpha-1 type I collagen.

## Data Availability

No new data were created or analyzed in this study. Data sharing is not applicable to this article.

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
