# Peer review of "Exploring the Role of Bempedoic Acid in Metabolic Dysfunction Associated Steatotic Liver Disease: Actual Evidence and Future Perspectives"

_ijms, 2024, doi:10.3390/ijms25136938_

Round 1
Reviewer 1 Report
Comments and Suggestions for Authors
In the present review, Butera et al. studied the promise held by bempedoic acid in blocking the progression of metabolic dysfunction-associated steatotic liver disease (MASLD) to metabolic dysfunction-associated steatohepatitis (MASH). Bempedoic acid has been approved for lowering cholesterol levels. As cholesterol also plays a role in the pathophysiology of MASLD and MASH, bempedoic acid could also have a favorable impact on this condition.
The manuscript is clearly written and its conclusions are supported by the literature. This work should be useful for researchers and clinicians working in the field of MASLD/MASH, an extremely prevalent condition.
Major remark:
The recently published article "Bempedoic acid for nonalcoholic fatty liver disease: evidence and mechanisms of action" (PMID: 36942869) appears to reach a similar conclusion to the present manuscript. Please highlight the novelty of the present manuscript, relative to this article.
Specific remarks:
1. Under Table 1, please include a footnote with the full names of the acronyms used in this table.
2. Due to redundancy, please consider replacing "lipotoxic lipids" with "toxic lipids".
3. In line 70, please include the full name of PCSK9.
4. In line 87, please include the full names of PNPLA3, TM6SF2, and Apo-B. The full name of PNPLA3 may be further removed from lines 303-304.
5. In lines 89-100, 105-110, 155-158, 159-161, 175-179, 183-185, 202-204, 307-309, and 369-373, please include supporting literature references.
6. In line 155, while mentioning statins, please consider mentioning their specific effects against cholesterol.
7. In lines 338-340, please include a reference for the statement "Finally, as half of individuals with MAFLD also have a related metabolic comorbidity (including 69% with hyperlipidemia, 51% with obesity, 39% with hypertension, 22% with type 2 diabetes, and 42% with metabolic syndrome), (...)".
8. In Table 2, please consider moving the abrreviations' list to a footnote under the table.
9. In Table 2, please use the same verb tense throughout.
10. In line 365, please correct "between" to "among", since it refers to a multiple comparison.
Author Response
Major remark:
The recently published article "Bempedoic acid for nonalcoholic fatty liver disease: evidence and mechanisms of action" (PMID: 36942869) appears to reach a similar conclusion to the present manuscript. Please highlight the novelty of the present manuscript, relative to this article.
Response: We managed to include 2 more recent studies in our review, and we tried to analyze the effect of BemA on MASLD compared to other lipid-lowering drugs.
Specific remarks:
- Under Table 1, please include a footnote with the full names of the acronyms used in this table.
Response: Thanks for the suggestion. However, we deleted Table 1 under suggestion of Reviewer n°2 as he felt that it was out of the main topic.
- Due to redundancy, please consider replacing "lipotoxic lipids" with "toxic lipids".
Response: we replaced the terms as suggested. Thanks for the tip.
- In line 70, please include the full name of PCSK9.
Response: Included.
- In line 87, please include the full names of PNPLA3, TM6SF2, and Apo-B. The full name of PNPLA3 may be further removed from lines 303-304.
Response: Included.
- In lines 89-100, 105-110, 155-158, 159-161, 175-179, 183-185, 202-204, 307-309, and 369-373, please include supporting literature references.
Response: We included references as asked.
- In line 155, while mentioning statins, please consider mentioning their specific effects against cholesterol.
Response: Thank you, we have added a detail on the mechanism of action of statins on cholesterol.
- In lines 338-340, please include a reference for the statement "Finally, as half of individuals with MAFLD also have a related metabolic comorbidity (including 69% with hyperlipidemia, 51% with obesity, 39% with hypertension, 22% with type 2 diabetes, and 42% with metabolic syndrome), (...)".
Response: Thanks for your suggestion, we added a precise reference to the statement.
- In Table 2, please consider moving the abrreviations' list to a footnote under the table.
Response: Thanks for the tip. We moved the abbreviations’ list to a footnote.
- In Table 2, please use the same verb tense throughout.
Response: We modified the table accordingly.
10. In line 365, please correct "between" to "among", since it refers to a multiple comparison.
Response: Thanks for the tip. Correction applied.
Reviewer 2 Report
Comments and Suggestions for Authors
In this review, Butera et al. summarize the roles of bempedoic acid in MASLD. This review is well-written and organized, and I have a few minor comments for this manuscript.
· As the title says, the main topic of this review is bempedoic acid, which is a medication for hypercholesterolemia. I feel like that some parts of this review, such as Table 1 showing risk factors for MASLD diagnosis, are out of this main topic. Table 1 is probably not needed. The authors should reconsider the structure of this manuscript and focus on the main topic. The authors describe statin and PCSK9 inhibitors as lipid-lowering agents, but if the authors want to discuss other medications, other lipid-lowering agents should also be discussed, such as PPAR agonists or fibrates. The authors also should discuss the advantages and disadvantages of bempedoic acid compared to other lipid-lowering agents. Several clinical trials are ongoing for fibrates in MASLD patients, and what is the reason for bempedoic acid to be used as another drug candidate for MASLD?
· The authors should discuss both preclinical or basic studies and clinical trials in this review. Table 2 lacks information that these studies are basic or clinical, and clinical trial ID, phase, patient numbers, current status, and reference numbers should be included in Table 2.
Author Response
Comment: As the title says, the main topic of this review is bempedoic acid, which is a medication for hypercholesterolemia. I feel like that some parts of this review, such as Table 1 showing risk factors for MASLD diagnosis, are out of this main topic. Table 1 is probably not needed.
Response: As suggested, we deleted Table 1 as it was not entirely focused on the main topic.
Comment: The authors should reconsider the structure of this manuscript and focus on the main topic. The authors describe statin and PCSK9 inhibitors as lipid-lowering agents, but if the authors want to discuss other medications, other lipid-lowering agents should also be discussed, such as PPAR agonists or fibrates. The authors also should discuss the advantages and disadvantages of bempedoic acid compared to other lipid-lowering agents.
Response: We added a brief description of the effects of PPAR agonist and fibrates on MASLD in order to compare advantages and disadvantages of BemA compared to other lipid-lowering agents.
Comment: Several clinical trials are ongoing for fibrates in MASLD patients, and what is the reason for bempedoic acid to be used as another drug candidate for MASLD?
Response: We believe that BemA could be useful in addition to statins for MASLD patients that don’t reach LDL target and in which the addition of fibrates could lead to myopathies. Indeed, the selective effect of BemA on liver enzymes makes it a valuable tool in MASLD patients with statin induced myopathies.
Comment: The authors should discuss both preclinical or basic studies and clinical trials in this review. Table 2 lacks information that these studies are basic or clinical, and clinical trial ID, phase, patient numbers, current status, and reference numbers should be included in Table 2.
Response: Thanks for the suggestion. We managed to include the ongoing randomized trial “Effect of Bempedoic Acid on Liver Fat in Individuals With Nonalcoholic Fatty Liver Disease and Type 2 Diabetes (B-LIFT)” NCT06035874 retrieved from ClinicalTrials.gov. Based on the research of ongoing clinical trials on ClinicalTrials.gov, to the best of our knowledge we were only able to retrieve NCT06035874. Since Table 2 contains only basic studies on murine models, we did not add NCT06035874 in Table 2. You can find a description of this trial in section “Bempedoic Acid and MASLD”.
As suggested, we included reference numbers in Table 2.
Reviewer 3 Report
Comments and Suggestions for Authors
The manuscript entitled "Exploring the role of bempedoic acid in metabolic dysfunction-associated steatotic liver disease: actual evidence and future perspectives" presents a comprehensive and well-structured overview of the potential use of bempedoic acid in treating metabolic dysfunction-associated steatotic liver disease (MASLD). The authors provide an in-depth exploration of the disease pathogenesis and the molecular mechanisms influenced by bempedoic acid in both human and animal models. The content is likely to be of significant interest to both basic and clinical researchers specializing in MASLD and liver diseases. The paper is suitable for publication with some minor revisions. Below are specific comments and suggestions for improvement:
Administration, Metabolism, and Stability of Bempedoic Acid:
It would enhance the paper to include a section on the pharmacokinetics of bempedoic acid, detailing its administration, metabolism, excretion, stability, and solubility. This information is crucial for understanding the practical application of the compound in clinical settings.
Impact of ACLY Inhibition on Citrate Levels:
Given that inhibition of adenosine-triphosphate citrate lyase (ACLY) may lead to increased citrate levels in the cytosol, it would be valuable to discuss potential consequences of this increase. Consider including whether there is a compensatory mechanism that decreases citrate efflux from the mitochondria or any other metabolic adjustments that may occur.
Cardiovascular Benefits:
Although the primary focus is on hepatic effects, cardiovascular events are a major concern for patients with MASLD. If there are studies indicating cardiovascular benefits of bempedoic acid, summarizing these findings would provide a more holistic view of the compound’s therapeutic potential.
Formatting of Tables and Figures:
The format of the tables can be improved for better readability and presentation. Consider refining the layout and design.
The size of Figure 1 should be reduced to fit better within the text and enhance readability.
In Figure 1, differentiate the shapes and colors of “Cholesterol crystals” and “Oxidized-low-density lipoproteins” to avoid confusion for readers.
Reference for Specific Statement:
Provide a reference for the statement on Page 3, lines 112-114 to ensure proper citation and support for the claim made.
Clarification on Ezetimibe:
Correct the title on Page 6, line 152 to include cholesterol absorption inhibitors like ezetimibe, distinguishing it from statin medications and PCSK9 inhibitors.
Consistency in Abbreviations:
Ensure consistent use of the abbreviation “HSC” throughout the manuscript, including Page 6, lines 177 and 178, and Page 9, lines 276.
Clarification on Liver Enzymes:
On Page 6, lines 185-188, specify which liver enzymes are significantly lowered by ezetimibe according to the referenced meta-analysis.
AMPK Regulation by Bempedoic Acid:
On Page 7, lines 185-188, clarify whether bempedoic acid upregulates the expression levels of AMP-activated protein kinase (AMPK) or increases its activity.
Correction in Figure 2:
In Figure 2, revise the representation near the term “diabetes” to ensure it accurately reflects the effects of bempedoic acid, as it does not directly reduce or decrease diabetes.
Correction in Table 2:
Correct the surname “Velàzquez” to “Velázquez” in Table 2 to ensure accuracy.
By addressing these points, the manuscript will be further strengthened, providing a clearer and more comprehensive presentation of the role of bempedoic acid in treating MASLD.

Author Response
Comment: Administration, Metabolism, and Stability of Bempedoic Acid:
It would enhance the paper to include a section on the pharmacokinetics of bempedoic acid, detailing its administration, metabolism, excretion, stability, and solubility. This information is crucial for understanding the practical application of the compound in clinical settings.
Response: Thank you for the valuable suggestion. We have added a detailed section on the pharmacokinetics of bempedoic acid, confident that this will contribute to the completeness of the review.
Comment: Impact of ACLY Inhibition on Citrate Levels:
Given that inhibition of adenosine-triphosphate citrate lyase (ACLY) may lead to increased citrate levels in the cytosol, it would be valuable to discuss potential consequences of this increase. Consider including whether there is a compensatory mechanism that decreases citrate efflux from the mitochondria or any other metabolic adjustments that may occur.
Response: That's an interesting point. We don't have a definitive answer on this matter. It's likely that there is metabolic crosstalk between different cellular compartments, and that the amount of intracellular citrate modulates the uptake of citrate from the extracellular space and the intramitochondrial space (through the citrate transporters SLC25A1 and SLC13A5). In any case, given the specificity of bempedoic acid, ACLY inhibition occurs only at the hepatic level and not systemically. (Fernandez-Fuente G, Rigby MJ, Puglielli L. Intracellular Citrate/acetyl-CoA flux and endoplasmic reticulum acetylation: Connectivity is the answer. Mol Metab. 2023 Jan;67:101653. doi: 10.1016/j.molmet.2022.101653. Epub 2022 Dec 10. PMID: 36513219; PMCID: PMC9792894.)
Comment: Cardiovascular Benefits:
Although the primary focus is on hepatic effects, cardiovascular events are a major concern for patients with MASLD. If there are studies indicating cardiovascular benefits of bempedoic acid, summarizing these findings would provide a more holistic view of the compound’s therapeutic potential.
Response: Thank you for the advice. Given the connection between MASLD and cardiovascular events, we have included a paragraph on the clinical trial that demonstrated the cardiovascular benefits of bempedoic acid, as suggested.
Comment: Formatting of Tables and Figures:
The format of the tables can be improved for better readability and presentation. Consider refining the layout and design.
Response: We modified the layout of Table 2. Under suggestion of Reviewer n°2 we deleted Table 1 as it was not entirely focused on the main topic. Therefore Table 2 became Table 1.
Comment: The size of Figure 1 should be reduced to fit better within the text and enhance readability.
Response: Thanks for the tip. We modified the Figure accordingly.
Comment: In Figure 1, differentiate the shapes and colors of “Cholesterol crystals” and “Oxidized-low-density lipoproteins” to avoid confusion for readers.
Response: Thanks for the tip. We modified the Figure accordingly.
Comment: Reference for Specific Statement:
Provide a reference for the statement on Page 3, lines 112-114 to ensure proper citation and support for the claim made.
Response: Thanks for the suggestion. We provided a reference as suggested.
Comment: Clarification on Ezetimibe:
Correct the title on Page 6, line 152 to include cholesterol absorption inhibitors like ezetimibe, distinguishing it from statin medications and PCSK9 inhibitors.
Response: Thank you for the suggestion. We corrected the title in “Lipid-lowering drugs and MASLD” to include ezetimibe and to specify that we focus on the effect on MASLD.
Comment: Consistency in Abbreviations:
Ensure consistent use of the abbreviation “HSC” throughout the manuscript, including Page 6, lines 177 and 178, and Page 9, lines 276.
Response: We modified the text accordingly.
Comment: Clarification on Liver Enzymes:
On Page 6, lines 185-188, specify which liver enzymes are significantly lowered by ezetimibe according to the referenced meta-analysis.
Response: Thank you for the suggestion. We have specified that it involves AST, ALT, and gamma-GT.
Comment: AMPK Regulation by Bempedoic Acid:
On Page 7, lines 185-188, clarify whether bempedoic acid upregulates the expression levels of AMP-activated protein kinase (AMPK) or increases its activity.
Response: Thank you for the advice. We have specified that it involves upregulation of the expression levels of AMPK, avoiding any confounding elements in the text and the image caption.
Comment: Correction in Figure 2:
In Figure 2, revise the representation near the term “diabetes” to ensure it accurately reflects the effects of bempedoic acid, as it does not directly reduce or decrease diabetes.
Response: We modified the Figure accordingly.
Comment: Correction in Table 2:
Correct the surname “Velàzquez” to “Velázquez” in Table 2 to ensure accuracy.
Response: Name corrected.